# HIV-1 re-suppression on a first-line regimen despite the presence of phenotypic drug resistance

Adriaan E. Basson[1,2]*, Salome Charalambous[3,4], Christopher J. Hoffmann[3,5], Lynn Morris[1,2]

1 Centre for HIV and STIs, National Institute for Communicable Diseases of The National Health Laboratory Services, Johannesburg, Gauteng, South Africa, 2 School of Pathology, Faculty of Health Sciences, University of the Witwatersrand, Johannesburg, Gauteng, South Africa, 3 The Aurum Institute, Johannesburg, Gauteng, South Africa, 4 School of Public Health, Faculty of Health Sciences, University of The Witwatersrand, Johannesburg, Gauteng, South Africa, 5 Johns Hopkins University, School of Medicine, Baltimore, Maryland, United States of America

* adriaan.basson@wits.ac.za

**Data Availability Statement:** All relevant data are within the paper and its Supporting Information files.

## Abstract

We have previously reported on HIV-1 infected patients who fail anti-retroviral therapy but manage to re-suppress without a regimen change despite harbouring major drug resistance mutations. Here we explore phenotypic drug resistance in such patients in order to better understand this phenomenon. Patients (n = 71) failing a non-nucleoside reverse transcriptase inhibitor (NNRTI)-based regimen, but who subsequently re-suppressed on the same regimen, were assessed for HIV-1 genotypic drug resistance through Sanger sequencing. A subset (n = 23) of these samples, as well as genotypically matched samples from patients who did not re-suppress (n = 19), were further assessed for phenotypic drug resistance in an *in vitro* single cycle assay. Half of the patients (n = 36/71, 51%) harboured genotypic drug resistance, with M184V (n = 18/36, 50%) and K103N (n = 16/36, 44%) being the most prevalent mutations. No significant difference in the median time to re-suppression (31–39 weeks) were observed for either group (p = 0.41). However, re-suppressors with mutant virus rebounded significantly earlier than those with wild-type virus (16 vs. 33 weeks; p = 0.014). Similar phenotypic drug resistance profiles were observed between patients who re-suppressed and patients who failed to re-suppress. While most remained susceptible to stavudine (d4T) and zidovudine (AZT), both groups showed a reduced susceptibility to 3TC and NNRTIs. HIV-1 infected patients on an NNRTI-based regimen can achieve viral re-suppression on the same regimen despite harbouring viruses with genotypic and phenotypic drug resistance. However, re-suppression was less durable in those with resistance, reinforcing the importance of appropriate regimen choices, ongoing viral load monitoring and adherence counselling.

## Introduction

The use of antiretroviral therapy (ART) has a significant impact on the control of HIV-1 infection and HIV associated morbidity [1]. In eastern and southern Africa, home to more than 19

**Funding:** AEB and LM received research grants from the National Health Laboratory Service (NHLS) and the Poliomyelitis Research Foundation (PRF) to fund this study.

**Competing interests:** The authors have declared that no competing interests exist.

million people living with HIV, new HIV infections have declined by nearly a third between 2010 and 2017, with a 42% decline in AIDS-related deaths [2]. South Africa's national anti-retroviral treatment (ART) program was rolled out in 2004 and is currently the largest treatment program in the world with ~4.3 million people receiving ART by 2017 [3].

The South African national guidelines for the management of HIV infection promote viral load testing for monitoring viral suppression on ART, as well as for diagnosing treatment failure [4]. At the time of this study, patients were initiated on an NNRTI-based first-line regimen and monitored by an annual viral load (VL) test. After two consecutive VL tests >1,000 copies/ml, patients were switched to a protease inhibitor (PI)-based second-line regimen. Patients failing an NNRTI-based first-line regimen with genotypic drug resistance mutations typically present with M184V/I, K65R, and/or thymidine analogue mutations (TAMs) and K103N, V106M/A and/or Y181C as the most prevalent NRTI and NNRTI mutations, respectively [5]. However, several studies have observed that 16%-71% of patients with viral breakthrough are able to re-suppress on the same NNRTI-based regimen after adherence intensification [6–13]. Re-suppression can be long-lasting and achievable for up to a median of 2.4 years [14].

Re-suppression has been observed in patients with major genotypic drug resistance mutations particularly those failing an NNRTI-based first-line regimen with M184V and K103N [7, 11, 13, 15]. The presence of such genotypic drug resistance mutations is usually associated with a decrease in the effectiveness of ART. However, the impact of these resistance mutations in the scenario of re-suppression is less clear. Here, we report on the genotypic resistance profiles of patients who re-suppressed on the same regimen, and performed *in vitro* phenotypic resistance testing to evaluate genotypic drug resistance in the context of re-suppression and failure.

## Materials and methods

### Study cohort

ART-naive patients were recruited and enrolled in a previously described workplace ART program within the mining industry, between November 2002 and May 2006, and initiated on an NNRTI-based combination ART regimen [13, 16]. At the time of initiation of ART, patients were offered the opportunity to participate in the evaluation cohort. Patients with CD4 count <250 cells/mm$^3$; WHO stage 3 and CD4 count <350 cells/ mm$^3$; or WHO stage 4, who initiated ART, were included in the overall cohort. CD4 counts and HIV RNA levels (VL) were determined before initiation, after 6 weeks on ART and every 6 months thereafter. Patients with a detectable viral load of >1,000 copies/ml after previously being undetectable, a sustained increase in viral load of >0.6 log from its lowest point and a return to 50% of the pre-treatment viral load were eligible to switch to a PI-based second line regimen. Due to concerns regarding adherence, tolerability of the second-line regimen, and premature regimen switching many patients where not switched to a second-line regimen until multiple elevated viral loads and, often, a notable CD4 count decline [17]. The patient demographics and study characteristics have been reported elsewhere [13]. Briefly, 93% (n = 3,479/3,727) of patients who met the inclusion criteria were male with a median age of 42 years and median CD4 count of 147 cells/mm3 at cART initiation. The median follow-up was 17.4 months, with 6,118 person-years of follow-up. The current study expands on previous studies from this cohort [13, 14] by including a larger number of samples for drug resistance genotyping, and is representative of the larger population of patients on NNRTI-based first-line ART regimens in South Africa. Patients with virologic failure and resuppression were included in this study based on the availability of stored serum for resistance testing. In addition, phenotypic drug resistance testing is performed on a subset of samples with a focus on resuppression and failure among individuals with genotypic HIV drug resistance. Ethics approvals were provided by the University

of the Witwatersrand Human Research Ethics Committee and Johns Hopkins University, and participants provided written consent.

## Genotypic drug resistance analysis

Total nucleic acid was extracted from patient plasma on the MagNA Pure LC 2.0 instrument (Roche, Switzerland). Viral RNA was reverse transcribed with the ThermoScript™ RT-PCR system (Invitrogen™, USA) and a ~1,150 kb HIV-1 reverse transcriptase (RT) fragment (from amino acid 82 in PR, to amino acid 367 in RT), amplified by nested PCR using the Expand High-Fidelity[PLUS] PCR System (Roche, Switzerland). The second-round primers introduced HpaI endonuclease restrictions sites on either side of the amplicons, as required for vector construction in the phenotypic assay [18]. Amplicons were sequenced by population-based sequencing using the ABI BigDye® Terminator V3.1 Cycle Sequencing Kit and the ABI Prism 3130xl Genetic Analyser (Applied Biosystems, USA). The Stanford HIV Drug Resistance Algorithm was used for quality assessment of sequences and mutation recognition [19].

## Phenotypic drug susceptibility analysis

The *in vitro* phenotypic drug susceptibilities of a subset of samples, containing viral variants with various genotypic drug resistance patterns, were assessed in a single-cycle non-replicative assay as previously described [18, 20]. Briefly, patient-derived pseudoviruses were constructed by ligating HpaI restricted nested PCR products into the p8.MJ4ΔRT HIV-1 subtype C expression vector. The patient-derived vectors included the RT connection/RNaseH domains to incorporate mutations (e.g. N348I [21]) that may impact on ARV drug susceptibility. The sequences of the inserts of the resulting clones were confirmed by population-based sequencing as described above. Clones whose genotypes matched the corresponding amplicons sequences were used. When genotypes did not match, more than one clone was used for that patient to ensure that all mutations were included. Pseudoviruses were generated by co-transfection of the patient-specific clones with vectors pM.DG and pCSFLW into HEK293T cells, and harvesting virus-containing supernatants 48 hours later. HEK293T cells were obtained from the American Type Culture Collection (Catalogue number CRL-3216) through Duke University School of Medicine. This cell line was not verified by our laboratory.

For the phenotypic analysis of patient-derived pseudoviruses, serial dilutions of anti-retroviral drugs were prepared in complete DMEM in flat-bottom 96-well culture plates. After the addition of HEK293T cells and pseudovirus, the plates were incubated for 48 hours and viral activity quantified through the expression of firefly luciferase using the Bright-Glo luciferase assay substrate (Promega, USA) on a Victor3 multi-label reader (PerkinElmer, USA). The p8. MJ4 expression vector was used as a wild-type control. Virus inocula were standardized to produce a luminescence of $1 \times 10^4$ to $1 \times 10^5$ relative light units (RLU) after 48 h of incubation with HEK293T cells in drug-free medium. The fold-change (FC) difference in inhibitory concentration-50 ($IC_{50}$) relative to the wild-type control was calculated for each patient-derived pseudovirus. The lower technical cut-off (TCO) for each drug was determined using the 99th percentile of the average $IC_{50}$ for the wild-type pseudovirus, assessed in two or more independent screens of each drug. The TCOs were as follows: AZT (2.1FC); d4T (2.9FC); ABC (1.1FC); 3TC (2.7FC); FTC (1.5FC); TDF (2.0FC); EFV (2.9FC); NVP (2.0FC). Samples with FC values at or below the TCO for a particular drug were classified as being susceptible, while those with FC values above the TCO were classified as having a reduced susceptibility. The TCO cut-off values used in this analysis were not linked to clinical correlates or outcomes but were used merely to rank the responses for the ARVs used in our assay.

## Drug level testing of EFV and NVP

Plasma samples from patients with resistance mutations who re-suppressed (n = 19) or failed (n = 16) were submitted to the National Health Laboratory Services (NHLS) for drug level testing by ultra-performance liquid chromatography–tandem quadruple mass spectrometry (UPLC-MS/MS) using a Waters Acquity UPLC T3 Column. For each sample, the protein in 25 μl of plasma was precipitated with 200 μl acetonitrile and 5 μl of deuterated internal standard. After centrifugation for 10 minutes at 14,000 x g, 4 μl of the precipitate was analysed using a gradient of $dH_2O$-acetonitrile (9:1) containing 0.1% formic acid. The limits of detection (LOD) for 3TC, EFV and NVP were 50.9 ng/ml, 285.7 ng/ml and 15.4 ng/ml, respectively.

## Calculations and statistical analysis

For the comparative analysis of patients characteristics between different groups, the unpaired t-test with Welch's correction factor (two-tailed) and contingency analyses with Fisher's exact test (two-sided, confidence interval (CI): 95%) were performed using GraphPad Prism version 5.08 for Windows (GraphPad Software, San Diego California USA, www.graphpad.com). For the *in vitro* phenotypic assay, $IC_{50}$ and FC values were determined using Microsoft Excel 2010 (Microsoft, Redmond, Washington, USA).

## Results

### Genotypic drug resistance analysis of patients who re-suppressed

For this study, 71 patients who showed an initial viral suppression (VL<400), followed by a viral breakthrough and a subsequent viral re-suppression on the same regimen were selected. The majority of patients were on a d4T/3TC (53/71 [75%]) or AZT/3TC (15/71 [21%]) containing regimen receiving either EFV (41/68) or NVP (27/68) The remaining 4% (3/71) of patients were on a regimen of TDF/FTC/EFV, TDF/FTC/NVP or ABC/3TC/EFV, respectively.

Almost all patients (70/71 [99%]) were infected with HIV-1 subtype C. One patient was infected with an HIV-1 CRF02_AG recombinant strain. Genotypic resistance analysis indicated that half of the patients (36/71 [51%]) contained virus with major genotypic drug resistance mutations (Table 1): NNRTI mutations (34/36 [94%]), NRTI mutations (18/36 [50%]) or both (16/36 [44%]). The most prevalent NNRTI mutations were K103N (16/36 [44%]) and V106M (9/36 [25%]). The prevalence of other major NNRTI resistance mutations (Y181CS, Y188CH, G190A and M230L) was ≤8%. The most predominant NRTI mutation was M184V (18/36 [50%]). Thymidine analogue mutations (TAMs) were present in four patients (11%), receiving either d4T (2/4), AZT (1/4) or ABC (1/4) (S1 Table). A quarter (9/36 [25%]) of the samples contained both M184V and K103N mutations.

Patients with mutant virus had a significantly (p = 0.005) higher median VL (2.31 $log_{10}$ copies/ml; IQR 1.39–3.32) during the full monitoring period compared to patients with wild-type virus (1.39 $log_{10}$ copies/ml; IQR 1.39–1.93). However, at the sampling time point, viral loads were significantly (p = 0.017) lower among those with mutant viruses (3.89 $log_{10}$ copies/ml vs. 4.35 $log_{10}$ copies/ml). Treatment responses after the sampling point showed that patients with mutant viruses had lower rates of virological re-suppression (to VL<400: 23 [64%]), higher rates of transient viremia (VL 400–1,000: 2 [6%]) and more repeat failures (VL>1,000: 11 [31%]) than those with wild-type virus (27 [77%]; 1 [3%]; 7 [20%], respectively).

Patients with mutant viruses were monitored for significantly (p = 0.020) longer periods (178 weeks, IQR 156–237) than those with wild-type virus (157 weeks, IQR 133–193). No significant difference (p = 0.508) was observed in the median time between visits for those with

**Table 1. Characteristics of 71 patients who re-suppressed with or without genotypic resistance.**

| Re-suppressor patient characteristics | [a]Wild-type virus n(%) | Mutant virus n(%) | |
|---|---|---|---|
| **Total** | **35** | **36** | |
| d4T/3TC + EFV | 18 (51) | 15 (42) | |
| d4T/3TC + NVP | 12 (34) | 8 (22) | |
| AZT/3TC + EFV | 2 (6) | 6 (17) | |
| AZT/3TC + NVP | 2 (6) | 5 (14) | |
| TDF/FTC + EFV | - | 1 (3) | |
| TDF/FTC + NVP | 1 (3) | - | |
| ABC/3TC + EFV | - | 1 (3) | |
| **Genotypes** | | **36** | |
| **All NNRTI** | | **34 (94)** | |
| K103N | | 16 (44) | |
| V106M | | 9 (25) | |
| Y181CS | | 3 (8) | |
| Y188H | | 3 (8) | |
| G190A | | 1 (3) | |
| M230L | | 1 (3) | |
| **All NRTI** | | **18 (50)** | |
| M184V | | 18 (50) | |
| TAMs | | 4 (11) | |
| **All NRTI + All NNRTI** | | **16 (44)** | |
| K103N+M184V | | 9 (25) | |
| **Viral load (log$_{10}$ copies/ml)** | *median (IQR)* | *median (IQR)* | *p-value* |
| During full monitoring period | 1.39 (1.78) | 2.31 (3.31) | 0.005 ** |
| At sampling point | 4.35 (4.96) | 3.89 (4.29) | 0.017 * |
| **Type of suppression after sampling point** | **35** | **36** | |
| Full suppression | 27 (77) | 23 (64) | |
| Transient viremia | 1 (3) | 2 (6) | |
| Repeat failure | 7 (20) | 11 (31) | |
| **Time (weeks)** | *median (IQR)* | *median (IQR)* | *p-value* |
| Full monitoring period | 157 (61) | 178 (80) | 0.020 * |
| Between visits | 26 (4) | 26 (8) | 0.508 ns |
| At sampling point (breakthrough) | 113 (90) | 125 (75) | 0.047 * |
| Re-suppressed | 39 (56) | 31 (45) | 0.408 ns |
| To repeat failure | 33 (32) | 16 (13) | 0.014 * |

IQR, Inter-Quartile Range; SD, Standard Deviation; ns, not significant;

* significant;

** very significant.

[a]Three patients (PID337454, PID120314 and PID316947) with polymorphisms (A98S, K103R, E138S, V179I) in the absence of major mutations were classified as having a wild-type genotype.

wild-type (26 weeks; IQR 24–29) and mutant virus (26 weeks; IQR 21–29). Sampling time points were significantly (p = 0.047) later for those with mutant virus (125 weeks; IQR 89–164) than for those with wild-type virus (113 weeks; IQR 52–142). No significant difference (p = 0.408) was observed in the median duration of re-suppression until repeat failure between those with wild-type (39 weeks; IQR 21–76) and mutant (31 weeks; IQR 25–63) virus. The maximum time of re-suppression was 116 and 98 weeks for wild-type and mutant samples,

respectively (not shown). For patients with a subsequent repeat failure, a significant difference (p = 0.014) was observed in time to the repeat failure for those with wild-type (median 33 weeks; IQR 24–57) and mutant (median 16 weeks; IQR 13–26) virus.

## Phenotypic drug resistance analysis of patients who re-suppressed or failed

From the 36 patients who re-supressed on the same regimen in the presence of one or more major NRTI and/or NNRTI mutations, 21 patients with various degrees of genotypic drug resistance (i.e. single, double and more complex combinations of major drug resistance mutations) representative of the cohort were selected for further phenotypic analysis. As a comparator group, 18 patients with similar drug resistance mutation profiles, who did not re-suppress or who had an in-class drug switch before re-suppression, were included (termed failing patients). For both groups, genotypic resistance patterns varied with some patients having single mutations and others with more complex combinations of up to 9 mutations (S2 Table). Two re-suppressor patients and one failure patient with wild-type genotypes were included as phenotypically susceptible controls.

When comparing the re-suppressor subgroup (n = 21) to the failing patients (n = 18), both groups had a median time between HIV RNA assays of ~24 weeks (IQR 20–27 weeks) and were followed over a similar time period of 193–243 weeks (Table 2). However, samples from failing patients were obtained at significantly (p = 0.006) later median time points (177 weeks; IQR 130–233 weeks) than for those who re-suppressed (153 weeks; IQR 100–163 weeks). Failing patients had significantly (p = 0.033) higher median viral loads (VL = 3.40 $\log_{10}$ copies/ml; IQR 2.98–3.98) during the entire monitoring period than those who re-suppressed (VL = 2.85 $\log_{10}$ copies/ml; IQR 1.99–3.4). Higher median viral loads were significantly associated with failure (OR 4.67, 95% CI 1.14–19.08, p = 0.049). There was no significant difference (p = 0.944) in the median viral loads at the sampling points. A significantly (p<0.0001) higher number of patients who failed (n = 17, 94%) had breakthrough events before the sampling point, compared to patients who re-suppressed (n = 4, 19%). We found that the number of previous breakthrough events were significantly predictive of failure (OR = 72.25, 95% CI 7.30–715.4, p<0.0001). Both groups of patients were on similar regimens and displayed similar genotypic resistance profiles, with the K103N (≥57%) and M184V (≥71%) mutations being the most prevalent.

A total of 43 pseudoviruses were generated from both groups and tested for *in vitro* phenotypic drug susceptibility against the ARVs that matched the regimen of the corresponding patients (S2 Table). The majority of cloned RT fragments had a genotype that agreed with their corresponding amplicon; 3 samples (PID486741, PID433732 and PID067414) had clones that showed genotypic variation and more than one clone was used in these cases. When comparing the fold-change in phenotypic susceptibility for each drug between the patients who re-suppressed or failed (Fig 1), the average FC values between the two groups of patients were generally comparable and not significantly different (p≥0.122). Although the average fold change for AZT was higher in patients who failed (FC = 19.9, SD 45.8) than for patients who re-suppressed (FC = 4.6, SD 7.4) this was not significantly different (p = 0.465). Almost all pseudoviruses (41/43 [95%]) showed high-level reductions in susceptibility to EFV or NVP (FC 4.7–54.1). Those with M184V mutation (33/43 [77%]) had a reduced susceptibility to 3TC (FC = 9.8, the limit of the assay), while those without M184V were fully susceptible (10/43 [23%], FC≤1.5) (S2 Table). A minority of pseudoviruses were resistant to d4T (3/23 [13%]) or AZT (5/15 [33%]). Three wild-type pseudoviruses were included as controls and were fully susceptible to all drugs. A subset of patient plasma samples were screened for detectable levels of 3TC, EFV and NVP as a proxy for ART compliance. A total of 19 and 16 samples from

**Table 2. Patients with genotypic resistance who re-suppressed compared to those who failed.**

| Patient characteristics | Re-suppressed n(%) | Failed n(%) | |
|---|---|---|---|
| Time (weeks) | *median (IQR)* | *median (IQR)* | *p-value* |
| Total follow-up period | 193 (79) | 243 (120) | 0.172 ns |
| Between visits | 25 (8) | 24 (6) | 0.272 ns |
| At sampling point | 153 (64) | 177 (103) | 0.006 ** |
| Viral load ($\log_{10}$ copies/ml) | | | |
| During follow-up | 2.85 (3.39) | 3.40 (3.93) | 0.033 * |
| At sampling point | 3.83 (4.37) | 4.05 (4.56) | 0.944 ns |
| Breakthrough events | | | |
| Before sampling | n = 4/21 (19%) | n = 17/18 (94%) | <0.0001 *** |
| **Total** | **21** | **18** | |
| d4T/3TC + EFV | 5 (24) | 4 (22) | |
| d4T/3TC + NVP | 6 (29) | 8 (44) | |
| AZT/3TC + EFV | 5 (24) | 5 (28) | |
| AZT/3TC + NVP | 4 (19) | 1 (6) | |
| TDF/FTC + EFV | 1 (5) | - | |
| **Genotype** | **21** | **18** | |
| **All NNRTI** | **20 (95)** | **18 (100)** | |
| K103N | 12 (57) | 11 (61) | |
| V106M | 4 (19) | 5 (28) | |
| Y181CS | 2 (10) | 1 (6) | |
| Y188CHL | 3 (14) | 3 (17) | |
| G190A | 1 (5) | 2 (11) | |
| M230L | 1 (5) | 1 (6) | |
| **All NRTI** | **15 (71)** | **14 (78)** | |
| M184V | 15 (71) | 14 (78) | |
| TAMs | 4 (19) | 5 (28) | |
| K65R | 1 (5) | 1 (6) | |
| **K103N+M184V** | **9 (43)** | **7 (39)** | |

IQR, Inter-Quartile Range; SD, Standard Deviation; ns, not significant;

* significant;

** very significant;

*** extremely significant

patients with genotypic resistance who re-suppressed or failed were available for drug level testing, respectively. A total of 63% (12/19) and 69% (11/16) of re-suppressor and failure patients, respectively, had detectable drug levels and this was not predictive of either re-suppressing or failing (OR 0.779, 95% CI 0.190–3.19, p = 1.000) (S2 Table).

The longitudinal profiles of patients with detectable drug levels are shown in Fig 2, together with their major drug resistance mutations and phenotypic susceptibilities. All 12 resuppressor patients showed control of viremia after the breakthrough event. Some patients had single mutations, mostly to NNRTIs while others had multiple mutations against both NNRTIs and NRTIs. There was a good correlation between the presence of genotypic resistance and an increase in fold-change to the corresponding ARV. This was observed despite the presence of detectable drug levels in all patients. Similar profiles were seen among the 11 failing patients except that viral levels remained high (>1,000) after the breakthrough event. One patient (PID

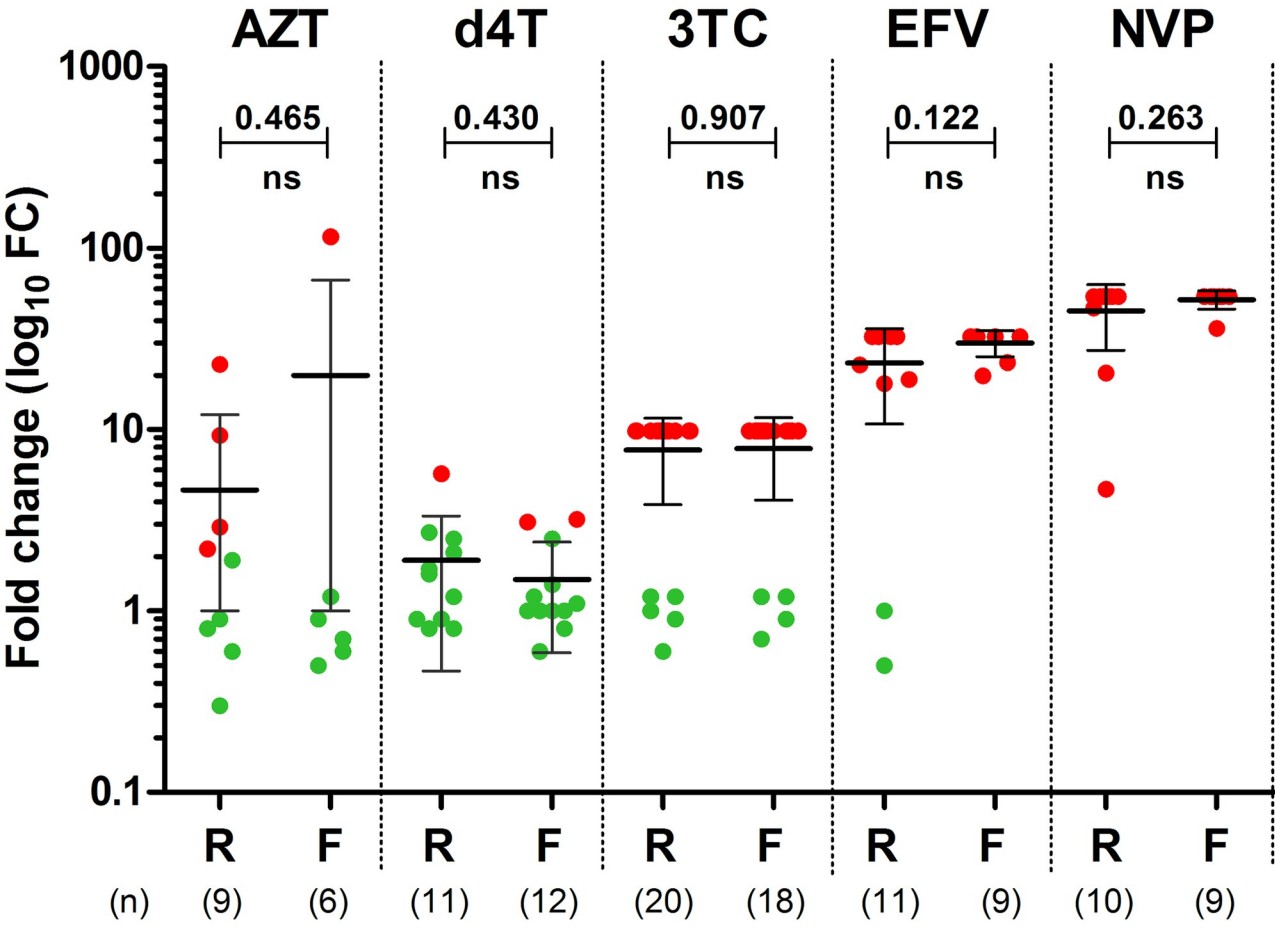

**Fig 1.** *In vitro* **phenotypic drug susceptibility of pseudoviruses from re-suppressing and failing patients.** The fold change (FC) in drug susceptibility of pseudoviruses from re-suppressing and failing patients for each antiretroviral drug used in their ARV regimens. The number of patients in each group is indicated below the x-axis. Averages, standard deviations (error bars) and statistical significances (p-values) are displayed. The cut-off for each drug is: AZT (FC<2.1), d4T (FC <2.9), 3TC (FC <2.7), EFV (2.9) and NVP (2.0). Pseudoviruses that are susceptible are shown in green and those with reduced susceptibility in red. ns–not significant. R: Patients who re-suppressed, F: Patients who failed.

438207) re-suppressed but only after an in-class drug switch. Thus, despite the similarities in detectable drug levels, the presence of genotypic drug resistance mutations and a reduced phenotypic susceptibility to one or more antiretroviral drugs in their regimen, some were able to resuppress while other patients failed treatment.

## Discussion

It is not uncommon for patients on ART that experience virological failure to re-suppress on the same regimen despite the presence of major drug resistance mutations that accumulated during sub-optimal ART. Here, we examined the impact of drug resistance mutations on the phenotypic response to ARVs in such patients from a workplace HIV programme that was initiated before roll-out of the national ARV treatment program in South Africa [13].

We found that half of the patients that re-suppressed after virologic breakthrough contained major NRTI (e.g. M184V) and NNRTI (e.g. K103N, V106M) drug resistance mutations, typically observed in patients failing an NNRTI-based regimen [22, 23]. These patients had more frequent repeat failures and lower viral loads at breakthrough events than patients that

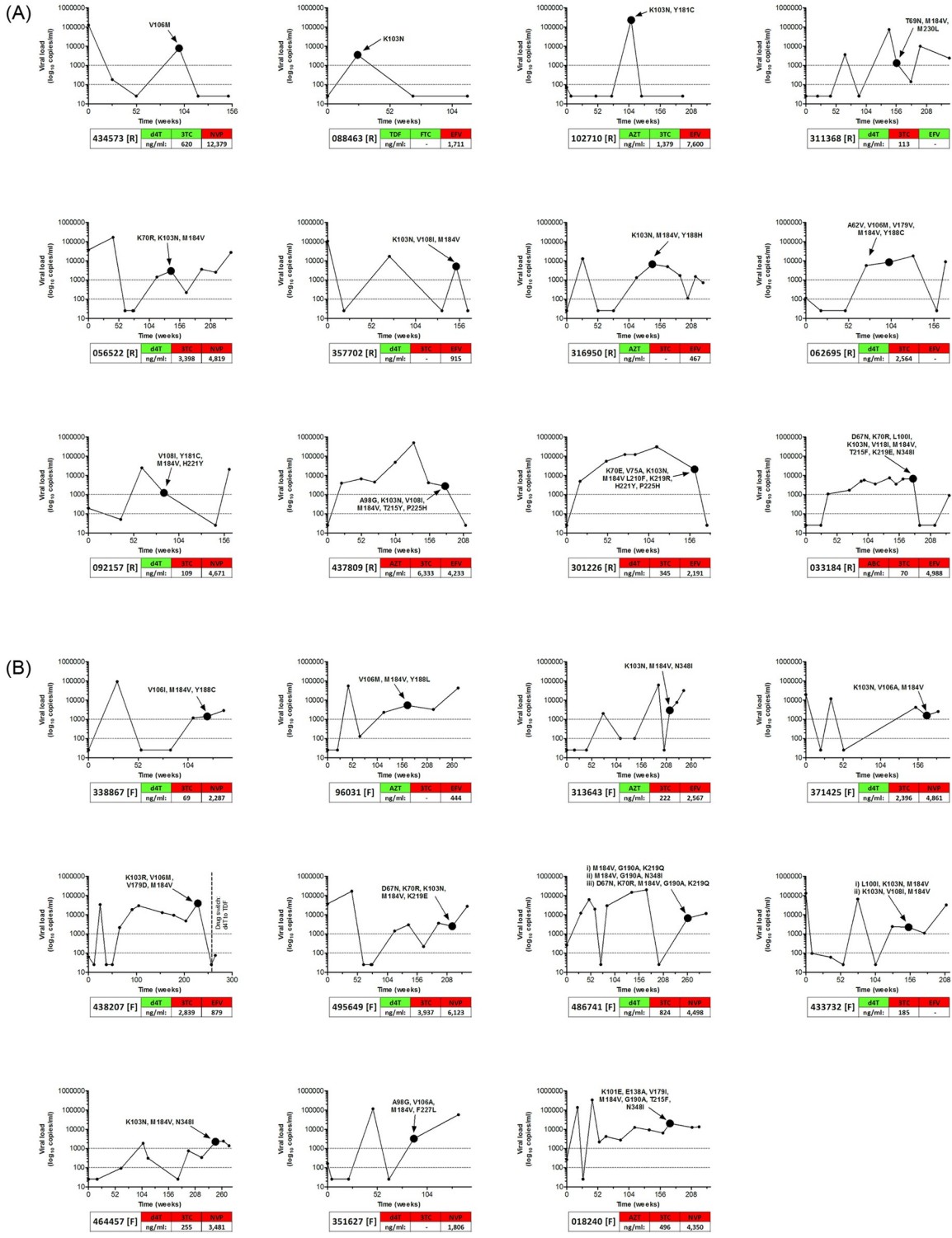

**Fig 2. Characteristics of patients with detectable drug levels, who re-suppressed or failed.** The viral loads of re-suppressing (A) and failing (B) patients are displayed over the monitoring period (in weeks). Sampling time points (large dot) and genotypic drug resistance mutations are shown. Drug regimens and *in vitro* phenotypic responses are indicated, with green indicating full susceptibility and red indicating a reduced susceptibility. Detectable drug levels are shown in ng/ml. Viral loads of 100 copies/ml and 1,000 copies/ml are indicated by dotted lines on the y-axis.

re-suppressed with wild-type virus. Overall, however, re-suppressors with drug resistance mutations had significantly higher viral loads during the treatment monitoring period. Although the median duration of re-suppression was similar between patients with or without genotypic resistance mutations (~7–9 months), those with resistance mutations were likely to have a repeat viral rebound much sooner, within a median of 17 weeks after a suppressed VL. An *in vitro* phenotypic drug susceptibility analysis between these patients and a comparator group of failing patients showed similar, predictable phenotypic resistance patterns. The majority of patients in both groups showed a decreased susceptibility to two antiretroviral drugs (3TC and EFVor NVP) while remaining fully or partially susceptible to only one antiretroviral drug (d4T or AZT). We found that the phenotypic response to ARVs was not predictable of failure, but that the number of previous viral breakthroughs were. These findings suggest that adherence, rather than viral susceptibility to a sub-optimal ART regimen, may be the determining factor with regards to re-suppression and failure.

It is well described that non-adherence is linked to the development of drug resistance and HIV disease progression [24]. Furthermore, a delayed antiretroviral switch is associated with an increased risk for the development of drug resistance, opportunistic infection (OI) and death [25, 26], and an increased risk of the transmission of drug resistant virus [27]. The relationship between adherence and the probability of drug resistance development differs dramatically across drug class [28, 29]. Resistance to NNRTI-based regimens occurs at low to moderate levels of adherence. Thus, in the case of re-suppressors with wild-type virus, it is likely that complete abstinence from cART occurred. In the case of re-suppressors with genotypic drug resistance mutations, low to intermediate adherence to the EFV/NVP-based regimen would have selected for the drug resistance mutations observed [28]. Despite the presence of genotypic resistance mutations, these patients were still able to re-suppress and we have shown no significant difference in phenotypic drug susceptibilities compared to patients who failed. For both the re-suppressor and failure groups, sensitivity to d4T or AZT remained, while loss in sensitivity to 3TC and EFV/NVP occurred. Although resistance to EFV/NVP is fairly absolute, NRTIs such as 3TC and AZT may have sustained ARV activity despite long intervals of partially suppressive therapy and the presence of resistant HIV-1 [30]. In addition, the M184V resistance mutation selected for by ABC, FTC or 3TC imparts a clinical benefit due to reduced viral fitness, increased RT fidelity, and hypersensitization to other NRTIs among others [31–33]. Despite this, however, drug resistance mutations selected for during sub-optimal ART may impact on future drug options and drug efficacies [34]. This is particularly relevant to NNRTI mutations and cross-resistance to newer second-generation NNRTIs (i.e. Etravirine and Rilpivirine) [18].

The detection of early virological failure should provide an opportunity to increase adherence counselling and repeat viral load testing before a switch in treatment [6]. Enhanced adherence counselling can result in the resuppression of a substantial number of patients failing cART and avoid unnecessary drug regimen switches [35]. Viral load monitoring has proven to be an important tool in monitoring and reinforcing adherence [8]. Same-day point-of-care viral load testing has shown to be useful in identifying patients who require immediate adherence councelling [36]. The inclusion of biomarkers and self-reported adherence could produce a potentially more accurate measure of adherence to cART [37]. Although therapeutic drug monitoring (TDM) can be useful in certain clinical settings, it should be done with caution. In the context of adherence monitoring, serum drug concentrations only reflect drug administration within a short time period prior to blood collection, and do not provide information on sustained drug dosing [38]. This is particularly true for drugs with short half-lives. Furthermore, ARV compliance in the days immediately before blood collection but lack of compliance otherwise (so-called "white coat syndrome") [39] could provide misleading conclusions in regard to adherence.

Non-adherence is multifactorial and solutions to circumvent this are complex [40]. Efforts should be intensified to identify patients at risk of poor adherence, and establish the support that is needed to overcome the most important barriers to adherence [41]. Enhanced adherence counselling alone, however, may not provide sufficient additional support than the current standard of care [42] and several successful interventions have been used to improve adherence and viral suppression [43, 44]. This includes regimen simplification [45], use of single-tablet regimens [46], short message service (SMS) text messages [47], and community-based ART programs [48, 49]. Although adherence intensification is likely to increase the probability of re-suppression on the same regimen, this does not always hold true [42, 50, 51]. Resistance testing, where feasible, has been proposed for those with higher levels of viremia at failure to improve discernment between patients in need of a switch to second-line therapy versus those in need of adherence support interventions [51]. Although genotypic drug resistance testing is not typically performed in low-to-middle income countries due to cost implications, alternative assays (e.g. point mutation assays) may provide a feasible option in the latter scenario [52]. Also, risk score algorithms may be useful in low-to-middle income countries to expedite drug or regimen switching for patients in need of more effective ART regimens [53].

There are some limitations in the current study. Firstly, only a limited number of patient samples were selected for phenotypic drug resistance testing, thereby potentially underpowering and limiting generalization of the results. Secondly, only a limited number of TDF-treated patients were included in this study and in light of the high prevalence of the K65R mutation in sub-Saharan African countries [54, 55], it is unclear whether our conclusions are applicable to patients re-suppressing or failing TDF-based regimens in this setting. Thirdly, in the absence of biological cut-off values for our *in vitro* phenotypic assay, an assumption was made that FC values above the TCO values translated into a decrease in drug susceptibility. This could potentially have lead to the under- or over-estimation of phenotypic drug resistance in the patient samples. Lastly, although we used drug level testing as a surrogate for adherence, no reliable adherence data was available for use in this study and could potentially limit the conclusions that were drawn. In addition, CD4+ T-cell counts and pre-treatment viral loads are important factors that could contribute to viral failure [56, 57] but were not considered in this study.

In conclusion, the ability of patients to re-suppress on the same regimen is possible despite genotypic resistance mutations and a reduced phenotypic susceptibility to most of the drugs in the regimen. However, the accumulation of additional genotypic drug resistance mutations in patients on sub-optimal ART regimens may have implication for future treatment options. Adherence is fundamentally important for a durable and sustained suppressive cART regimen. Monitoring and support is pivotal and should be provided sooner than later. Drug resistance testing may be beneficial in guiding the management of virologic failure for some individuals in resource-limited settings.

## Supporting information

**S1 Table. Characteristics and genotypic drug resistance mutations of re-suppressor patients.**
(XLSX)

**S2 Table. Characteristics, genotypic- and phenotypic drug resistance of a subset of resuppressing and failing patients.**
(XLSX)

## Acknowledgments

The following reagents were obtained through the NIH AIDS Reagent Program, Division of AIDS, NIAID, NIH: Abacavir (ABC), Efavirenz (EFV), Emtricitabine (FTC), Lamivudine (3TC), Stavudine (d4T), Nevirapine (NVP), Tenofovir disoproxil fumarate (TDF) and Zidovudine (AZT). Plasmids p8.9 and pMDG were supplied by Didier Tron (École Polytechnique Fédérale de Lausanne, Lausanne, Switzerland), and plasmid pCSFLW was supplied by Nigel Temperton (University College London, London, United Kingdom). The authors would like to thank Tracey Snyman and Derryn Legg-Esilva from the NHLS for performing the drug level testing.

## Author Contributions

**Conceptualization:** Christopher J. Hoffmann, Lynn Morris.

**Data curation:** Adriaan E. Basson.

**Formal analysis:** Adriaan E. Basson.

**Funding acquisition:** Adriaan E. Basson, Lynn Morris.

**Investigation:** Adriaan E. Basson, Christopher J. Hoffmann, Lynn Morris.

**Methodology:** Adriaan E. Basson.

**Project administration:** Lynn Morris.

**Resources:** Salome Charalambous, Christopher J. Hoffmann.

**Supervision:** Lynn Morris.

**Writing – original draft:** Adriaan E. Basson.

**Writing – review & editing:** Adriaan E. Basson, Salome Charalambous, Christopher J. Hoffmann, Lynn Morris.

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
