## [Decision Letter · Decision Letter 0]

1 May 2020

PONE-D-20-09445

HIV-1 re-suppression on a first-line regimen despite the presence of phenotypic drug resistance

PLOS ONE

Dear Dr. Basson,

Thank you for submitting your manuscript to PLOS ONE. After careful consideration by myself and one reviewer, we feel that it has merit but does not fully meet PLOS ONE’s publication criteria as it currently stands. Therefore, we invite you to submit a revised version of the manuscript that addresses the points raised during the review process.

Please address the comments from the Reviewer 1.

We would appreciate receiving your revised manuscript by Jun 15 2020 11:59PM. To enhance the reproducibility of your results, we recommend that if applicable you deposit your laboratory protocols in protocols.io, where a protocol can be assigned its own identifier (DOI) such that it can be cited independently in the future. For instructions see: http://journals.plos.org/plosone/s/submission-guidelines#loc-laboratory-protocols

We look forward to receiving your revised manuscript.

Kind regards,

Cristian Apetrei, MD, PhD

Academic Editor

PLOS ONE

Journal Requirements:

2. In your Methods section, please provide additional details regarding the cell lines used in your study. Please include the source from which you obtained the cells, whether the cell line was verified, and if so, how it was verified.

For more information on PLOS ONE's guidelines for research using cell lines, see

https://journals.plos.org/plosone/s/submission-guidelines#loc-cell-lines

3. To comply with PLOS ONE submission guidelines, in your Methods section, please provide additional information regarding your statistical analyses. Specifically, please indicate which of the described statistical analyses were performed for which experiment. For more information on PLOS ONE's expectations for statistical reporting, please see https://journals.plos.org/plosone/s/submission-guidelines.#loc-statistical-reporting.

4. Thank you for your ethics statement "ethics approvals were provided by the University of the Witwatersrand Human Research Ethics Committee and Johns Hopkins University, and participants provided written consent."

Please add this information to your methods section as well.

5. In your Methods section, please provide additional information about the participant recruitment method and the demographic details of your participants.

Please ensure you have provided sufficient details to replicate the analyses such as:

a) the recruitment date range (month and year),

b) a description of any inclusion/exclusion criteria that were applied to participant recruitment,

c) a table of relevant demographic details,

d) a statement as to whether your sample can be considered representative of a larger population, and

e) a description of how participants were recruited.

Reviewers' comments:

Reviewer's Responses to Questions

**Comments to the Author**

1. Is the manuscript technically sound, and do the data support the conclusions?

Reviewer #1: Yes

2. Has the statistical analysis been performed appropriately and rigorously? 

Reviewer #1: Yes

3. Have the authors made all data underlying the findings in their manuscript fully available?

Reviewer #1: Yes

4. Is the manuscript presented in an intelligible fashion and written in standard English?

Reviewer #1: Yes

5. Review Comments to the Author

Reviewer #1: The study titled “HIV-1 re-suppression on a first-line regimen despite the presence of phenotypic drug resistance” investigates the phenotypic drug resistance in HIV-1 infected patients who fail anti-retroviral therapy but re-suppress on the same regimen in spite of possessing drug resistance mutations. The study and the experiments were designed well, although the authors did not find any significant differences in the phenotype resistance patterns between the study group and the comparator group. The authors, however, report that the number of previous viral breakthroughs, and therefore, adherence to treatment regimen, may be the determining factor for re-suppression and failure. The manuscript was well-written.

Minor Comment

1. In lines 239 and 240, the authors have selected 21 patients for phenotypic analysis out of the 36 patients who were re-suppressed on the same regimen in the presence one of more major NRTI and/or NNRTI mutations. The authors have not explained why all the 36 patients were not considered for the phenotypic analysis or what criteria was used to select the 21 patients. Please elaborate.

6. PLOS authors have the option to publish the peer review history of their article (what does this mean?). If published, this will include your full peer review and any attached files.

Reviewer #1: No

---

## [Author Response · Author response to Decision Letter 0]

13 May 2020

Reviewer 1: Minor Comment

In lines 239 and 240, the authors have selected 21 patients for phenotypic analysis out of the 36 patients who were re-suppressed on the same regimen in the presence one of more major NRTI and/or NNRTI mutations. The authors have not explained why all the 36 patients were not considered for the phenotypic analysis or what criteria was used to select the 21 patients. Please elaborate.

Response: We have now clarified this in the text:

Page 12, lines 262 – 266: “From the 36 patients who re-supressed on the same regimen in the presence of one or more major NRTI and/or NNRTI mutations, 21 patients with various degrees of genotypic drug resistance (i.e. single, double and more complex combinations of major drug resistance mutations) representative of the cohort were selected for further phenotypic analysis.”

---

## [Editor Report · Decision Letter 1]

5 Jun 2020

HIV-1 re-suppression on a first-line regimen despite the presence of phenotypic drug resistance

PONE-D-20-09445R1

Dear Dr. Basson,

We’re pleased to inform you that your manuscript has been judged scientifically suitable for publication and will be formally accepted for publication once it meets all outstanding technical requirements.

Kind regards,

Cristian Apetrei, MD, PhD

Academic Editor

PLOS ONE
---

## [Editor Report · Acceptance letter]

9 Jun 2020

PONE-D-20-09445R1 

HIV-1 re-suppression on a first-line regimen despite the presence of phenotypic drug resistance 

Dear Dr. Basson:

I'm pleased to inform you that your manuscript has been deemed suitable for publication in PLOS ONE. Congratulations! Your manuscript is now with our production department. 

Kind regards, 

on behalf of

Dr. Cristian Apetrei 

Academic Editor

PLOS ONE